# SWIFTSAGE: A Generative Agent with Fast and Slow Thinking for Complex Interactive Tasks

**Bill Yuchen Lin** [1]    **Yicheng Fu** [4]    **Karina Yang** [2]    **Faeze Brahman** [13]    **Shiyu Huang** [5]
**Chandra Bhagavatula** [1]    **Prithviraj Ammanabrolu** [67]    **Yejin Choi** [31]    **Xiang Ren** [21]

[1]Allen Institute for Artificial Intelligence
[2]University of Southern California    [3]University of Washington    [4]Tsinghua University
[5]4Paradigm Inc.    [6]University of California, San Diego    [7]MosaicML

https://swiftsage.github.io

## Abstract

We introduce SWIFTSAGE, a novel agent framework inspired by the dual-process theory of human cognition, designed to excel in action planning for complex interactive reasoning tasks. SWIFTSAGE integrates the strengths of behavior cloning and prompting large language models (LLMs) to enhance task completion performance. The framework comprises two primary modules: the SWIFT module, representing fast and intuitive thinking, and the SAGE module, emulating deliberate thought processes. The SWIFT module is a small encoder-decoder LM fine-tuned on the oracle agent's action trajectories, while the SAGE module employs LLMs such as GPT-4 for subgoal planning and grounding. We develop a heuristic method to harmoniously integrate the two modules, resulting in a more efficient and robust problem-solving process. In 30 tasks from the ScienceWorld benchmark, SWIFTSAGE significantly outperforms other methods such as SayCan, ReAct, and Reflexion, demonstrating its effectiveness in solving complex interactive tasks.[1]

## 1 Introduction

The advancement of artificial general intelligence is largely dependent on the development of agents that are proficient in complex interactive reasoning tasks. These agents should be capable of exhibiting problem-solving abilities akin to humans within dynamic, open-world environments [26, 7]. For example, the ScienceWorld benchmark [36] features a task where an agent must determine the electrical conductivity of an unknown object. In a simulated environment, the agent must navigate to appropriate rooms, locate and acquire essential items, such as batteries and light bulbs, build a circuit, perform an experiment, and interpret the results. Tackling such a complex interactive task demands agents to exhibit long-horizon planning, long-term memorization, subgoal decomposition, spatial reasoning, exception handling, and commonsense knowledge capabilities [37].

There are three primary approaches to developing agents capable of addressing complex interactive reasoning tasks: (1) (deep) reinforcement learning (RL), (2) behavior cloning (BC) [34] through sequence-to-sequence (seq2seq) learning [33], and (3) prompting large language models (LLMs) [6]. In addition to conventional RL methods such as DRRN [14], interactive reasoning can be framed as a seq2seq task, where the input text serves as the current state description and the output text corresponds to the subsequent action [9, 3]. By leveraging numerous gold trajectories generated by oracle agents, it becomes feasible to fine-tune Transformer models [35], like T5 [25], to effectively imitate the

---

[1]Contact: yuchenl@allenai.org

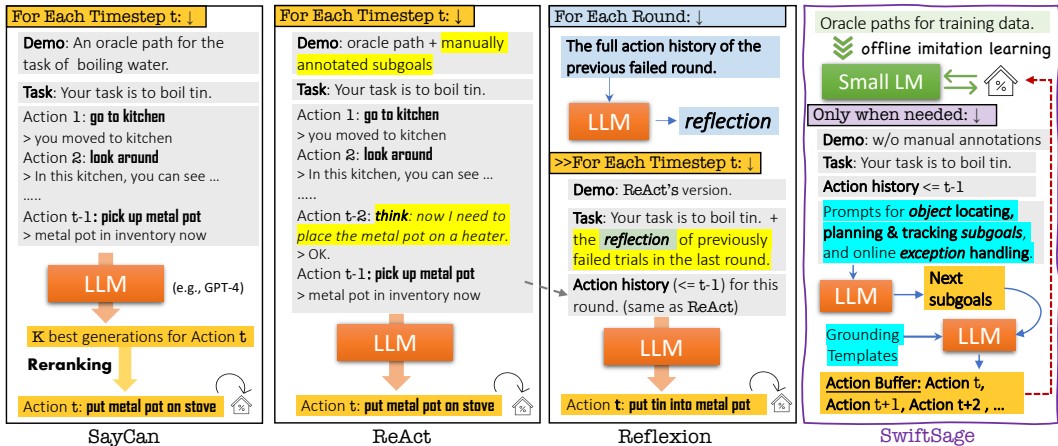

Figure 1: **Comparing methods of prompting LLMs to build agents for interactive tasks.**

behavior of these oracle agents. Recent studies have also demonstrated that generative agents based on prompting LLMs, such as GPT-4, can produce reasonable plans and actions [18, 15, 32].

Although the aforementioned methods exhibit remarkable performance in relatively simple tasks, their ability to generalize to more complex and demanding tasks is limited. Both RL-based and seq2seq-based BC approaches effectively acquire knowledge from the environment through large-scale interactions and learn general action patterns from oracle agents. However, they face difficulties in decomposing tasks into subgoals, maintaining long-term memory, generalizing to unseen tasks, and handling exceptions. In contrast, instruction-tuned LLMs [24] demonstrate the ability to generate reasonable high-level plans for complex tasks and adapt their outputs based on human feedback. Yet, grounding their outputs to executable actions in the environment remains a challenge. These procedures also lack the capability to efficiently handle environment-specific exceptions that prevent agents from adhering to the LLM's plans. Additionally, previous methods such as SAYCAN [1], REACT [41] and REFLEXION [30] require a new inference with LLMs for each time step, making them considerably costly and inefficient (see Figure 1).

Inspired by the dual process theory [39, 16], we propose a novel framework that enables agents to closely emulate how humans solve complex, open-world tasks. The dual-process theory posits that human cognition is composed of two distinct systems: System 1, characterized by rapid, intuitive, and automatic thinking; and System 2, which entails methodical, analytical, and deliberate thought processes. System 1 is reminiscent of seq2seq methods, which learn through imitation of oracle agents and primarily operate utilizing shallow action patterns. Conversely, System 2 bears resemblance to LLMs that excel in applying commonsense knowledge, engaging in step-by-step reasoning, devising subgoal strategies, and exercising self-reflection. Thus, our proposed method, SWIFTSAGE, is designed to enable both fast and slow thinking in complex interactive reasoning tasks. It effectively integrates the strengths of behavior cloning (representing System 1) and prompting LLMs (emulating System 2), resulting in significant enhancements in task completion performance and efficiency.

Specifically, SWIFTSAGE consists of two primary modules: the SWIFT module and the SAGE module. The SWIFT module is a small encoder-decoder LM, fine-tuned on a T5-large (770m) checkpoint using the searched oracle trajectories of training tasks. It encodes short-term memory components, such as previous actions, observations, visited locations, as well as the current environment state. Then, it decodes the next individual action. This module simulates the fast, intuitive thinking characteristic of System 1. The SAGE module, representing the deliberate thinking of System 2, utilizes LLMs, such as GPT-4, and is structured around two prompting stages: planning and grounding. In the planning stage, we prompt LLMs to locate necessary items, plan and track subgoals, as well as detect and fix potential exceptions and mistakes. In the grounding stage, we focus on utilizing LLMs to transform the output subgoals derived from the planning stage into a sequence of actions by demonstrating potential action templates. Unlike prior methods, where LLMs only generate the next immediate action, our procedures engage in longer-term action planning. To harmoniously integrate the SWIFT and SAGE modules, we developed a heuristic algorithm that determines when to (de)activate the SAGE module and how to combine the outputs effectively with an action buffer mechanism.

In a comprehensive evaluation on 30 task types from the ScienceWorld benchmark, SWIFTSAGE significantly outperforms other methods, achieving a state-of-the-art average score of 84.7. In comparison, SAYCAN scores 33.8, REACT obtains 36.4, and REFLEXION reaches 45.3. Moreover, SWIFTSAGE is more cost-effective and efficient, requiring much fewer tokens per action for LLM inference than previous methods. This considerable performance advantage highlights the effectiveness and efficiency of the SWIFTSAGE framework in addressing complex interactive tasks.

## 2 Background and Related Work

### 2.1 Complex Interactive Reasoning

We define interactive reasoning as the problems where agents are tasked with accomplishing a goal within an interactive environment, typically simulated by engines such as AI2Thor [17] and TextWorld [11]. Our focus lies on the textual environment of ScienceWorld [36] and the complex interactive tasks it supports. Simple interactive tasks, like those created in ALFWorld [31] and TWC [21], primarily involve searching for and placing objects as well as performing basic actions within a single location. Many of these simple tasks have been almost solved by recent works.

In contrast, tasks in **ScienceWorld** exhibit greater complexity, characterized by more challenging task planning and a significantly larger action space (encompassing 10 locations, 200 types of objects with varying states, and 25 types of actions). Furthermore, agents may encounter random, unforeseen obstacles, such as broken stoves or missing soil, which hinder the execution of planned actions. As a result, agents must adapt and re-plan accordingly, for example, by seeking alternative heat sources or using a shovel on the outside ground to get soil. These challenges demand that agents possess skills in long-horizon planning, long-term memory, subgoal decomposition, exception handling, and commonsense knowledge—capabilities that are not explicitly required for simple interactive tasks.

### 2.2 Reinforcement Learning and Imitation Learning Methods

**DRRN.** Interactive tasks can naturally be framed as partially-observable Markov decision processes (POMDPs), enabling the application of RL-based methods. Deep Reinforced Relevance Network (DRRN) [14] is a standard baseline method to learn agents within text environment. It aims to learn representations of observations and actions separately and train a policy network to select actions from candidates based on feedback from the simulated environment. **CALM** [40] is a reranking-based method that combines DRRN with a causal language model (LM) fine-tuned with oracle transcripts. In essence, the causal LM captures task-specific and environment-specific knowledge through imitation learning, and the DRRN learns to rerank the predictions from the LM.

The **KG-A2C** [2] method uses an OpenIE technique [4] to represent environment states with graph structures and dynamically update these graphs. These graphs guide policy networks by constraining the combinations of action templates and objects. This method has been shown to be effective in other domains such as for multimodal embodied agents [22].

**Behavior cloning for offline imitation learning.** Behavior cloning is an imitation learning method that trains a seq2seq Transformer offline with action transcripts of similar training tasks generated by oracle agents [34, 3]. During training, it uses the previous action, observation at time step $t - 1$, and the current observation as input and learns to output the next action. The Text Decision Transformer (**TDT**) is a textual variant of the Decision Transformer [9], which also employs behavior cloning and uses the same data. The primary innovation of TDT is the introduction of reward-to-go as part of the inputs, enabling the model to learn predicting actions that maximize future expected rewards.

### 2.3 Prompting LLMs for Action Planning.

Language models (LLMs) such as GPT-4 have shown promise for action planning in interactive tasks [18, 15, 32, 38]. In this paper, we adapt three prominent methods to complex interactive reasoning tasks in ScienceWorld: SAYCAN [1], REACT [41], and REFLEXION [30].

**SAYCAN** [1] is a straightforward agent that integrates an LLM with a value function of underlying policies regarding grounding affordances (i.e., the feasibility of an action in the environment). We

need to provide the history and current environment as textual inputs to LLMs for generating a ranked list of action candidates. This action list is then reranked based on a value function.

**REACT** [41] presents a virtual 'think' action, enabling LLMs to generate *subgoals* during action planning. This approach requires human annotators to supply examples of correct subgoals for each task type, employing few-shot in-context learning to teach LLMs *when* and *how* to 'think' in order to plan subsequent subgoals, in addition to providing complete action trajectories.

**REFLEXION** [30], a recent work building on REACT, proposes a multi-round approach enabling LLMs to use the history of previously failed rounds to refine their planning for the next round. This self-reflection mechanism helps LLMs improve after each failed attempt. However, this may not be practical in real-world applications for many tasks, as actions in failed trials can be irrecoverable.

All three methods require a new LLM inference at each time step to predict the next immediate action, resulting in inefficient and costly agents. REACT and REFLEXION require human annotations of correct subgoals for each unseen task type. Moreover, it is difficult to generalize REFLEXION to real-world situations where trial-and-error approaches can be infeasible for embodied tasks.

### 2.4 Dual-Process Theory

The dual-process theory [39, 16] is a cognitive psychological framework proposing the existence of a fast and a slow thinking systems in the human brain. This influential theory has found widespread applications across various fields, highlighting the critical role of both systems in shaping human cognition [5, 8, 12, 20, 23]. By integrating the complementary strengths of both systems, agents can effectively and efficiently handle diverse challenges in real-world scenarios. Inspired by this, we aim to construct a generative agent that utilizes a small seq2seq LM as System 1 for associative reasoning via behavior cloning while developing System 2 for analytical reasoning by prompting LLMs.

## 3  SWIFTSAGE: A Generative Agent with Fast and Slow Thinking

In this section, we first establish the problem. Then, we present the two core modules, SWIFT and SAGE, individually. Lastly, we demonstrate the integration of these two modules. , resulting in a harmonious and effective interactive reasoning process.

### 3.1  Problem Formulation

**Environment and tasks.**   We focus on complex interactive reasoning tasks situated in virtual textual environments such as ScienceWorld [36]. ScienceWorld provides an optimal setting for developing and evaluating agents in *complex* tasks, comprising 30 distinct task types covering 10 topics in science experiments. It features 10 locations, including an art studio, workshop, kitchen, living room, bedroom, bathroom, foundry, greenhouse, outdoor area, and a connecting hallway. The environment includes 200+ object types with multiple states (e.g., open, activated) and supports 25 action templates, resulting in an intractable search space. The simulator can generate numerous variations of each task type, providing a rich training ground. In each variation, the agent and environment initialization, such as the locations and states of objects, will differ. A plethora of training variations encompassing all task types are available for training agents. Additionally, it provides a handcrafted oracle agent to search for successful transcripts with minimal actions for offline learning.

Evaluation is done on a set of testing variations with unseen combinations of required objects and situations, thus substantially different from the training variations. For example, a training variation may involve boiling water, while a testing variation could require boiling tin. Therefore, it is crucial to ensure the agent's compositional generalization ability for effectively handling real-world scenarios.

**Interactions.**   Given a task variation, an agent is provided with the task description $D$ and the initial environment state ($t = 0$). The task description $D$ is a text specifying a high-level goal, e.g., "*Your task is to test if an unknown substance A is electronically conductive.*" At each time step $t$, the agent generates an action $A_t$ based on a set of supported action templates (e.g., pick up X, use X on Y). $A_0$ is always "look around" for showing initial environment information. Upon receiving an action from the agent, the environment produces feedback in four dimensions:

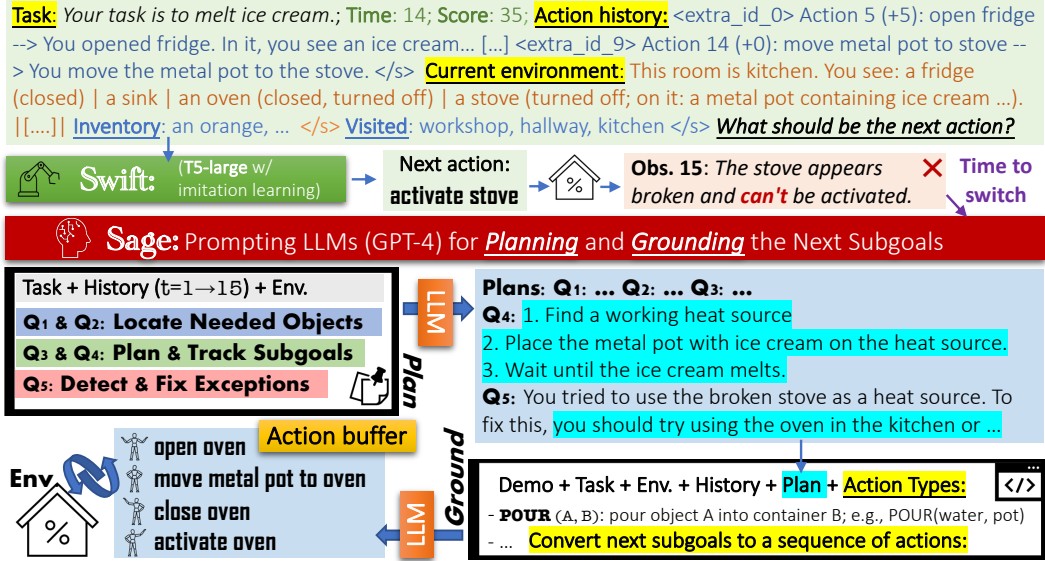

Figure 2: **An example of how SWIFTSAGE works with fast and slow thinking.** The SWIFT module is offline trained via imitation learning with a small LM such as T5-large (770m). When it is necessary, for example, encountering an exception, we switch to the SAGE module that prompts LLMs (e.g., GPT-4) for planning and grounding the next subgoals, resulting in an action buffer.

- **Observation** $O_t$ provides direct feedback on the action $A_t$ regarding its effects on the environment or the information queried. For example, an $A_t$ of "`use thermometer on the substance in metal pot`" may result in an $O_t$ like "*The temperature is 80F.*"
- **Environment** $E_t$ represents the current room in which the agent is situated and provides details about all visible objects. Object visibility is based on container states, e.g., objects within a closed fridge are not included in $E_t$ until the agent performs an action like "`open fridge.`"
- **Inventory** $I_t$ lists objects picked up by the agent, which is particularly useful when agents collect items from different locations to complete the task.
- **Score** $S_t$ represents the agent's cumulative score ranging from 0 to 100. When a required intermediate state is achieved, the score increases with a positive reward.

## 3.2 SWIFT: The Module for Intuitive and Associative Thinking via Imitation Learning

Imitation learning is used to construct an agent that learns to mimic oracle agents in various training scenarios through seq2seq learning. Previous methods, such as TDT [36], mainly employ one-hop history as input context and learn to output the subsequent action $A_t$ [36]. However, these methods exhibit limitations due to their *restricted context* of action history and harmful biases arising from *data imbalance*. To address these issues, we introduce our SWIFT module, depicted in Figure 2.

**Representation for longer history.** We expand the conventional one-hop BC to multi-hop by incorporating a sliding window of observations and rewards for the $K = 10$ recent actions. Additionally, we include a special field for visited rooms (without duplication). This approach aims to provide agents with a longer context and prevent unnecessary room navigation. The input format is as follows: "`Task:` $D$; `Time:` $t-1$; `Score:` $S_{t-1}$; `Action history:` $[A_{t-i}$ $(+R_{t-i}) \rightarrow O_{t-i}]$ /\* $i$ loops from $K$ to 1\*/; `Current room:` $E_{t-1}$; `Inventory:` $I_{t-1}$; `Visited rooms:` $\{E_1^*, \ldots, E_{t-1}^*\}$". Here, $R_t = S_t - S_{t-1}$ represents the reward at $t$, and $E_t^*$ is the location name at $t$.

**Balanced imitation learning.** To avoid bias caused by data imbalance for seq2seq learning, we down-sampled specific types of tasks and actions to achieve a more balanced final dataset for training. We used the T5-large with 770 million parameter and instruction-following ability [10], creating an efficient agent that we named SWIFT. Our empirical results show that the SWIFT module performs much better than TDT (11 billion) despite being 15x smaller in size.

The SWIFT module exhibits greater accuracy during initial time steps, enabling it to attain higher scores in the early stages of a complex task. However, it often fails to generalize to unseen situations. The module also has a tendency to repeat meaningless actions when its learned plans yield exceptions from the environment (e.g., the broken stove in Figure 2). This is partly due to the nature of imitation learning, which prioritizes emulating the observable actions of oracle agents rather than their intrinsic planning abilities. Besides, since the oracle trajectories contain only the shortest, correct actions, it is thus also challenging for the SWIFT to learn how to fix mistaken actions.

### 3.3 SAGE: The Module for Deliberate and Analytical Thinking via Prompting LLMs

While the SWIFT module acquires surface knowledge about the environment and task types through imitation learning, it lacks two key abilities essential for complex interactive reasoning: 1) generalizable planning and tracking of subgoals, and 2) robust handling of exceptions. Prior research has shown that LLMs outperform smaller LMs in these abilities. They can perform step-by-step reasoning to devise concrete plans for tasks and self-refine their outcomes. However, the performance of prior methods remains unsatisfactory in complex interactive tasks such as those in ScienceWorld.

We introduce a novel two-stage approach, named SAGE. This method initially acquires higher-level recommendations from LLMs during the planning stage, followed by their translation into specific action sequences in the grounding stage. By decoupling the planning and grounding processes, SWIFTSAGE effectively generates a series of actions for completing the planned subgoals.

**Planning stage.** In this stage, we leverage LLMs to plan based on the current state. Specifically, we prompt LLMs with a single prompt that includes a summarized version of the task description and action history, and asks the following five key questions:

> ▶ **Q1** (locate objects): "*To complete the task, which objects do I need to collect? Please list them and their possible locations one by one.*"
> ▶ **Q2** (track objects): "*Are there any objects that have not been collected yet?*"
> ▶ **Q3** (plan subgoals): "*To complete the task most efficiently, what are the important subgoals to achieve? Please list the subgoals one by one.*"
> ▶ **Q4** (track progress): "*Considering these subgoals, what have I already completed? And which subgoal should I focus on right now?*"
> ▶ **Q5** (handle exceptions): "*Have I made any mistakes that might prevent me from efficiently completing the next subgoal? If any , how should I fix them?*"

Before posing the five planning-related questions, we condense the entire action history ($A_{<t}$ and $O_{<t}$), and the current environment information $E_{t-1}$. **Q1** and **Q2** pertain to objects, as acquiring all necessary objects serves as the foundation for effective task planning. By addressing these questions, we ensure that LLMs develop a comprehensive understanding of the current environment. **Q3** prompts LLMs to engage in step-by-step planning by decomposing the task into a series of subgoals. **Q4** acts as a follow-up question, allowing the agent to monitor its progress based on the action history and determine completed subgoals, subsequently focusing on the remaining tasks. Lastly, **Q5** is employed to identify and address potential exceptions. These questions can be further tailored with additional environment-specific hints, thereby enhancing their adaptability.

To improve the structure of the LLMs' outputs and facilitate parsing, we incorporate additional instructions in the prompt. By utilizing a *single* input to obtain answers to all five questions in one output, rather than engaging in multiple rounds of interactive prompting, our approach is more efficient and cost-effective than the iterative prompting methods.

**Q4** and **Q5** are of primary importance, while **Q1**−**Q3** serve as auxiliary guidance for the LLMs. If the action history indicates a mistaken action or an unachievable previous subgoal, the response to **Q5** refines the answer to **Q4** through *self-reflection on the fly*. This approach differs from the REFLEXION agent, which only prompts reflective questions at the end of a failed trial, allowing agents to improve their planning in subsequent attempts. In contrast, our method detects exceptions and errors each time the agent plans for the next subgoals, enabling earlier correction of the agent's behavior.

**Grounding stage.** While the answers to **Q1**−**Q5** provide valuable guidance for agents, they are not directly executable. Converting plans into valid actions that can be accepted by the environment

remains a challenge. Previous methods using LLMs over-generate candidates, and they rely on reranking or filtering based on the action space to select the next action. However, this is inefficient and inaccurate for complex tasks with vast action spaces. Additionally, these methods generate a single action at a time, which can be both costly and ineffective for long-horizon tasks.

To tackle these issues, we first present supported action types using a formal style accompanied by remarks. For instance, the action type "`pour X into Y`" is introduced as "`POUR(X, Y)`: *pour object X into container Y; e.g., pour red paint into wood cup*". More examples:

> `TELEPORT(room)` : directly go to a room such as `TELEPORT(kitchen)`
> `PICK(object)` : pick up an object and put it into your inventory
> `OPEN(object)` : open an object to search or put things in it, e.g., `OPEN(freezer)`.
> `ACTIVATE(object)` : activate / turn on an object such as sink or stove, so that you can use it.
> `DEACTIVATE(object)` : deactivate / turn off the object
> `EXAMINE(object)` : look at an object carefully. For example, `EXAMINE(light bulb)`.
> `MOVE(object, place)` : move/place the object to a place

We then incorporate the LLM's outputs from the planning stage as part of the input for the grounding stage. Furthermore, we provide the recent action history of the past 10 time steps as context. Finally, we prompt LLMs to concentrate on the next subgoal and convert it into a *list* of actions (rather than a single action) to accomplish the next subgoal. Our formatting instructions enable the straightforward splitting and conversion of output actions from LLMs in the grounding stage back to their original action representations. We denote this list of actions generated by LLMs as the *action buffer*: $B = \{\hat{A}_t, \hat{A}_{t+1}, \dots\}$. One can opt to use only answers to Q4 and Q5 to reduce computational costs. Our small-scale ablation study indicates that incorporating answers to Q1-Q3 in the grounding stage proves beneficial, yielding a gain of about 2 points for short tasks on average.

### 3.4 Integration of Fast and Slow Thinking

Having described the SWIFT and SAGE modules, we now address the question of how to merge both modules and effectively integrate fast and slow thinking within the SWIFTSAGE agent. We establish a heuristic algorithm to control the activation and deactivation of the two modules.

Initially, we employ the SWIFT module due to its superior intuitive reasoning capabilities, which facilitate accurate associations between the task description and the environment during the first few actions. We will switch from SWIFT mode to SAGE when any of the following conditions are met:

> 1) **Stuck**: There are K=5 consecutive time steps with zero reward ($\sum_{i=t-5}^{t-1} R_i = 0$).
> 2) **Invalid**: The SWIFT's prediction for the next action ($A'_t$) is invalid in the current environment.
> 3) **Critical**: $A'_t$ involves a critical decision, e.g., giving the final answer for the experiment result.
> 4) **Unexpected**: The observation of $A'_t$ suggests that an exception is encountered.

Upon activating the SAGE module, we execute the two-stage prompting process and generate an action buffer. We attempt to execute each predicted action and revert to the SWIFT module when the buffer is empty. This approach enables a seamless integration of both modules, providing an efficient and robust problem-solving process for the SWIFTSAGE agent. The pseudo code for illustrating the SwiftSage framework is shown in Fig. 4 (Appendix).

## 4 Evaluation

### 4.1 Evaluation Setup

To evaluate the effectiveness of SWIFTSAGE and other baseline methods in complex interactive reasoning tasks, we use the ScienceWorld benchmark. In Section 2.1 and Section 3.1, we introduce the benchmark and problem formulation. Each task type is categorized as 'short' (`S`), 'medium' (`M`), or 'long' (`L`) based on the average length of the oracle truth trajectories. However, the length of the task does not necessarily indicate its level of difficulty as some tasks may require additional commonsense knowledge. Further evaluation details are provided in the appendix.

Table 1:

| Task Type | *Len | DRRN | KGA2C | CALM | TDT | SayCan | ReAct | Reflexion | SwiftSage |
|---|---|---|---|---|---|---|---|---|---|
| **1**-1 (L) | 107.7 | 3.52 | 0.0 | 0.0 | 0.71 | 33.06 | 3.52 | 4.22 | 97.04 |
| 1-2 (L) | 78.6 | 3.52 | 0.0 | 0.0 | 0.44 | 10.39 | 13.70 | 10.61 | 87.04 |
| 1-3 (L) | 88.9 | 0.0 | 4.0 | 0.0 | 3.88 | 3.88 | 7.78 | 7.78 | 72.78 |
| 1-4 (L) | 75.2 | 0.0 | 0.0 | 0.0 | 0.55 | 0.37 | 9.88 | 0.92 | 100.0 |
| **2**-1 (M) | 21.4 | 6.56 | 6.0 | 1.0 | 6.16 | 26.37 | 7.19 | 5.92 | 99.17 |
| 2-2 (M) | 35.2 | 5.50 | 11.0 | 1.0 | 6.43 | 8.03 | 6.10 | 28.59 | 88.17 |
| 2-3 (L) | 65.0 | 6.0 | 4.0 | 1.0 | 19.87 | 17.41 | 22.37 | 22.37 | 95.73 |
| **3**-1 (S) | 13.6 | 12.0 | 7.0 | 5.0 | 40.55 | 52.14 | 56.0 | 100.0 | 88.67 |
| 3-2 (M) | 20.8 | 9.0 | 4.0 | 7.0 | 14.26 | 22.50 | 54.33 | 17.45 | 55.33 |
| 3-3 (M) | 25.6 | 9.05 | 4.0 | 2.0 | 10.16 | 99.56 | 76.19 | 72.54 | 71.90 |
| 3-4 (M) | 29.0 | 9.52 | 4.0 | 2.0 | 21.65 | 47.76 | 88.81 | 70.22 | 77.86 |
| **4**-1 (S) | 14.6 | 15.0 | 18.0 | 10.0 | 41.93 | 22.87 | 26.67 | 64.93 | 100.0 |
| 4-2 (S) | 8.8 | 45.0 | 44.0 | 54.0 | 55.76 | 58.18 | 80.0 | 87.27 | 100.0 |
| 4-3 (S) | 12.6 | 21.67 | 16.0 | 10.0 | 27.82 | 20.87 | 53.33 | 16.42 | 91.67 |
| 4-4 (S) | 14.6 | 19.17 | 15.0 | 8.0 | 47.15 | 31.43 | 27.50 | 100.0 | 100.0 |
| **5**-1 (L) | 69.5 | 8.0 | 6.0 | 2.0 | 6.89 | 9.92 | 9.06 | 7.33 | 74.59 |
| 5-2 (L) | 79.6 | 14.29 | 11.0 | 4.0 | 11.86 | 13.93 | 18.57 | 13.0 | 93.93 |
| **6**-1 (M) | 33.6 | 15.77 | 17.0 | 3.0 | 15.10 | 47.81 | 51.04 | 70.35 | 49.40 |
| 6-2 (S) | 15.1 | 26.67 | 19.0 | 6.0 | 15.70 | 39.26 | 58.89 | 70.67 | 100.0 |
| 6-3 (M) | 23.0 | 10.37 | 4.0 | 3.0 | 5.25 | 19.72 | 40.74 | 15.77 | 91.48 |
| **7**-1 (S) | 7.0 | 50.0 | 43.0 | 6.0 | 30.0 | 80.0 | 60.0 | 100.0 | 95.0 |
| 7-2 (S) | 7.0 | 50.0 | 32.0 | 10.0 | 8.43 | 67.50 | 67.50 | 84.37 | 85.0 |
| 7-3 (S) | 8.0 | 33.33 | 23.0 | 4.0 | 8.34 | 50.0 | 50.0 | 83.0 | 93.33 |
| **8**-1 (M) | 40.0 | 21.0 | 5.0 | 4.0 | 3.86 | 20.91 | 27.67 | 2.58 | 89.0 |
| 8-2 (S) | 16.3 | 8.0 | 10.0 | 0.0 | 8.0 | 16.0 | 8.0 | 8.0 | 68.50 |
| **9**-1 (L) | 97.0 | 10.0 | 4.0 | 0.0 | 2.53 | 21.94 | 40.50 | 50.63 | 75.0 |
| 9-2 (L) | 84.9 | 10.0 | 4.0 | 3.0 | 14.66 | 32.26 | 44.0 | 100.0 | 70.0 |
| 9-3 (L) | 123.1 | 10.0 | 4.0 | 2.0 | 9.12 | 13.67 | 41.0 | 70.62 | 60.0 |
| **10**-1 (L) | 130.1 | 16.80 | 11.0 | 2.0 | 1.51 | 67.53 | 25.70 | 50.90 | 92.30 |
| 10-2 (L) | 132.1 | 17.0 | 11.0 | 2.0 | 1.29 | 59.45 | 16.80 | 23.69 | 77.60 |
| Short | *11.76* | 28.08 | 22.70 | 11.30 | 28.37 | 43.83 | 48.79 | 71.47 | 92.22 |
| Medium | *28.58* | 10.85 | 6.88 | 2.88 | 10.36 | 36.58 | 44.01 | 35.43 | 77.79 |
| Long | *94.30* | 8.26 | 4.92 | 1.33 | 6.11 | 23.65 | 21.07 | 30.17 | 83.0 |
| **Overall** | *49.26* | **15.56** | **11.37** | **5.07** | **14.66** | **33.82** | **36.43** | **45.34** | **84.68** |

Table 1: **Results on the ScienceWorld benchmark.** *Len is the average length of the oracle agent's trajectories. In addition to overall results, we also report performance on three groups of *Len (short, medium, long). The last four methods use GPT-4 as the base LLM for prompting. We show more details of these tasks and results on other LLMs in the appendix.

## 4.2 Baseline Agents

In addition to the baseline methods evaluated in the ScienceWorld paper, such as DRRN, CALM, KG-A2C, and TDT, we incorporate three LLM-based prompting techniques: SAYCAN, REACT, and REFLEXION, as detailed in Section 2.3 and Figure 1. This subsection presents the implementation details for adapting these methods to build ScienceWorld agents.

SAYCAN necessitates a value function from the environment for reranking purposes. We employ SentenceBERT [27] to rank all valid actions (generated by ScienceWorld's APIs) based on their similarity to the top 5 generations for $A_t$ from SAYCAN. We implemented REACT and REFLEXION in a similar manner. Adhering to their released code, we utilized the best single generation and determined the valid action with the minimal edit distance, if required. Both REACT and REFLEXION necessitate subgoal annotations for teaching LLMs to plan with virtual 'think' actions. We annotated such truth subgoals by translating ScienceWorld's APIs into natural language, which was also employed by the oracle agents. For all agents, we incorporated the complete trajectories of one or two training variations from the same task type for in-context learning. Our primary experiments were conducted using OpenAI's GPT-4; however, other LLMs can be readily substituted as required.

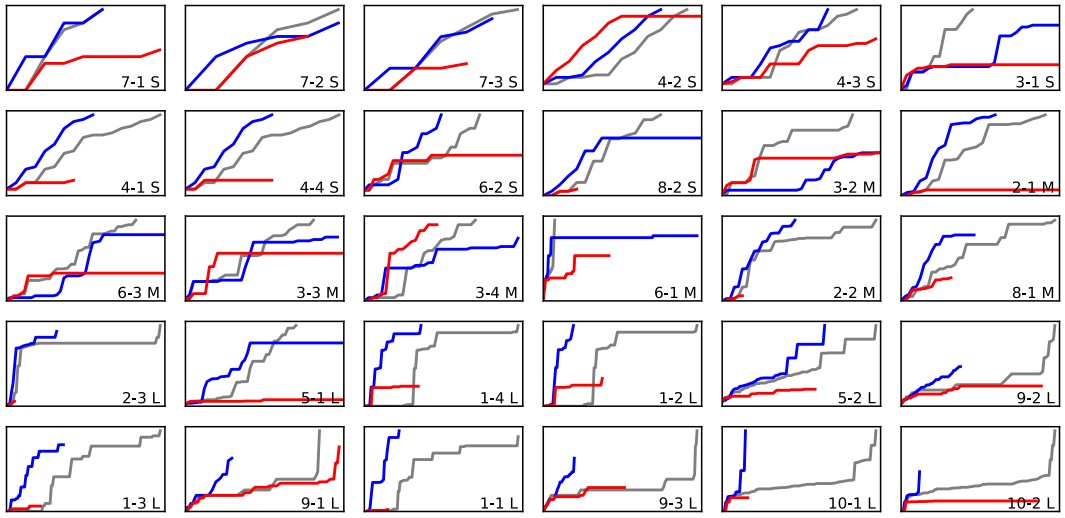

Figure 3: **Visualizing trajectories of SWIFTSAGE, REACT and ORACLE.** $X$: time steps ($0 \rightarrow T$); $Y$: scores ($0 \rightarrow 100$). Each figure displays the merged trajectories of testing variations by an agent in each task. Task IDs are shown at the bottom-right, and the ordering is based on *Len* in Tab 1.

### 4.3 Results and analysis.

**Main Results**  Table 1 compares the performance of various agents across 30 types of tasks. Detailed descriptions of each task type can be found in the ScienceWorld paper [36] and our appendix. It is evident that LLM-based methods outperform conventional agents due to their superior generalization ability, albeit at a higher deployment cost. The behavior cloning model TDT [36, 9] (11b) performs on par with DRRN [14], but with greater efficiency in learning and inference. In contrast, our SWIFT-only agent (770m) achieves an overall performance of 49.22, which we attribute to its balanced training data and the use of a sliding window for longer action histories.

REACT demonstrates a noticeable improvement over SAYCAN for short and medium tasks, owing to its subgoal annotations for in-context learning and the inclusion of 'think' actions. REFLEXION surpasses REACT in shorter tasks; however, comparing REFLEXION with other agents is not entirely fair. REFLEXION can run up to four rounds, while the others are limited to one round. This discrepancy is particularly unfair for tasks involving multiple-choice scenarios. Nevertheless, we include REFLEXION's results to analyze the potential of such methods.

**Exception handling.**  Consider the example in Figure 2, where the stove is broken, presenting an exception. Agents like DRRN and TDT often resort to repeating meaningless action sequences (e.g., continuously attempting to activate the stove or moving between rooms aimlessly). Although the SWIFT module, when used independently, improves upon this due to its larger context window from imitation learning, it still struggles to address exceptions robustly. ReAct and Reflexion occasionally utilize the 'think' action or reflections to redirect agents towards alternative solutions, but the generated actions rarely achieve the new subgoals if they are not grounded. In contrast, the plan-and-ground prompts in our SAGE module handle exceptions more effectively.

**Cost-effectiveness.**  Despite SAGE invoking LLMs APIs twice for inference, its overall cost remains lower, as the result is a *sequence* of actions typically containing about 5 actions. In comparison, SAYCAN and REACT require **1,855.84** and **1,971.03** *tokens per action* (tpa) respectively, while REFLEXION necessitates **2,983.46** tpa. SWIFTSAGE, on the other hand, only uses **757.07** tpa. Given its superior performance, SWIFTSAGE proves more cost-effective than other LLM-based methods. This efficiency is primarily attributed to invoking LLMs only when needed (courtesy of our strong SWIFT module) and the action buffer mechanism.

**Efficiency.**  To thoroughly examine the efficiency of agents across all task types, we use Figure 3 to visualize the average trajectories of the first three testing variations for each task involving

SWIFTSAGE, REACT, and the oracle agent. We arrange the tasks based on their average lengths of oracle trajectories (*Len* in Table 1). We observe that oracle trajectories consistently achieve perfect scores, yet SWIFTSAGE can reach similar scores more efficiently. This is particularly evident in longer tasks (the bottom two rows), although SWIFTSAGE does not achieve a perfect score for a few tasks (e.g., 9-2 and 1-3). Interestingly, we find that REACT performs competitively in shorter tasks (e.g., 4-2 and 3-4), but most trajectories plateau at an intermediate score and fail to reach 100.

**More analysis.** Due to page limit, we have to provide further details and analysis in the appendix, including more detailed analysis on cost-effectiveness and efficiency, additional case studies and ablation studies, sensitivity to LLM choices, and an the evaluation of the SWIFT-only agent.

## 5 Conclusion

**Contributions.** We present SWIFTSAGE, a generative agent for complex interactive reasoning tasks, inspired by the dual-process theory of human cognition. The agent framework comprises two modules: SWIFT, responsible for fast thinking, and SAGE, dedicated to slow thinking. The SWIFT module is a smaller LM that is fast and specialized, while the SAGE module focuses on prompting LLMs (e.g., GPT-4) for subgoal planning and reflective thinking. Through extensive experiments on 30 distinct tasks within the ScienceWorld benchmark, SWIFTSAGE outperforms baseline agents, achieving state-of-the-art performance, increased efficiency, and reduced cost.

**Implications.** The success of SWIFTSAGE highlights the potential for collaborative frameworks combining smaller LMs and LLMs in complex reasoning tasks. Smaller LMs can be trained more easily to recognize task-specific and environment-specific patterns, fostering effective in-distribution generalization. On the other hand, LLMs demonstrate remarkable zero-shot generalization abilities and deliberate thinking, though grounding their outputs in real-world environments remains challenging. We posit that dual-process agents, harnessing the strengths of both approaches, constitute a crucial step towards addressing complex interactive reasoning tasks and building general AI agents. Additionally, we can regard SWIFTSAGE as a method within the broader context of utilizing LLMs as controllers or planners for decomposing complex tasks and leveraging APIs/tools [19, 13, 29, 28]. To this end, we have explored applying SWIFTSAGE in web tasks and coding for math problems.

**Limitations.** Our work has been evaluated solely within a *textual* simulator, ScienceWorld, which supports a limited set of actions and tasks compared to real-world situations. Also, we did not implement any safeguards to prevent agents from engaging in potentially hazardous actions that could occur in the real world, such as picking up substances from a blast furnace. We argue that one important future direction is to develop a true open-ended environment, allowing agents to interact with a much wider variety of actions and objects to better emulate real-world scenarios. Besides, the use of LLMs in SAGE may present scalability challenges, as LLMs require significant computational resources and may not be feasible in some settings. Future research should explore the generalizability of SWIFTSAGE to other domains and the potential for more lightweight approaches to slow thinking. In addition, we believe it is important to train agents beyond simple supervised fine-tuning and to learn a trainable module to decide when to switch between SWIFT and SAGE mode.

## Acknowledgements

We thank Peter Jansen, Eric Xingdi Yuan, and Marc-Alexandre Côté for valuable discussions. We thank members of the INK lab at USC and the Mosaic team at AI2 for valuable feedback on this project. Xiang Ren is supported in part by the Office of the Director of National Intelligence (ODNI), Intelligence Advanced Research Projects Activity (IARPA), via the HIATUS Program contract #2022-22072200006, the DARPA MCS program under Contract No. N660011924033, the Defense Advanced Research Projects Agency with award W911NF-19-20271, NSF IIS 2048211, and gift awards from Google and Amazon. This research was also supported by the DARPA MCS program through NIWC Pacific (N66001-19-2-4031) and Allen Institute for AI. The views and conclusions contained herein are those of the authors and should not be interpreted as necessarily representing the official policies, either expressed or implied, of ODNI, IARPA, or the U.S. Government.

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
