# OpenReview forum: "SwiftSage: A Generative Agent with Fast and Slow Thinking for Complex Interactive Tasks"
_NeurIPS.cc/2023/Conference — NeurIPS 2023 spotlight_

### Official Review · Reviewer_MDXU · 2023-07-03

**Soundness:** 3 good
**Presentation:** 2 fair
**Contribution:** 3 good
**Rating:** 7
**Confidence:** 4

**Summary:**

The paper suggests an approach to solving interactive reasoning and planning tasks inspired by the dual-system theory of human cognition popularized by Kahneman. The approach, SwiftSage, combines a small seq2seq model as System 1 and a LLM as System 2. A heuristic module mitigates between the two. The authors conduct extensive experimentation on a large number of textual reasoning and planning tasks. SwiftSage achieves better performance than previous approaches by a large margin in terms of overall performance, efficiency, and exception handling.

**Strengths:**

The paper tackles an important and well-studied problem of solving interactive reasoning and planning tasks. The authors suggest a nice combination between small seq2seq LMs and LLMs, building on the strengths of each one on its own. While this idea is fairly simple, I consider it a solid contribution. Evaluation is extensive and convincingly demonstrates the superiority of the approach over previous ones in terms of performance, efficiency ad exception handling. Overall the paper is clear to read and well motivated.

**Weaknesses:**

There are some related works that should be mentioned and perhaps even included in the evaluation. One is about combining LLMs and classical planners [1]. The other is about a different LLM-based dual-system approach [2].

Statements about LLMs resembling System 2 in human cognition (lines 50-52) are overblown in my opinion. While LLMs sometimes exhibit a behavior that may seem like deliberate reasoning, some may argue that System 2 in [2] is more akin to a humans’ System 2. I suggest to water down such statements or at least address the controversy.

[1] https://escholarship.org/content/qt3qq6w5kx/qt3qq6w5kx_noSplash_2e490248a98936f442c743ad768b65f4.pdf
[2] https://openreview.net/forum?id=uyKk_avJ-p4


**Questions:**

Lines 24-26 – are those the only approaches? What about using classical planners as part of the solution (e.g. [1] above).

Line 205 – I thought that the two stages of planning+grounding are both part of the Sage module. Here you refer to them as SwiftSage.

Heuristic for switching between Swift and Sage. – what are the critical decisions (other than giving the final answer)?

Sec. 4.3 – how much of the improvement is attributed to a balanced training set? Has this been checked on other models?



**Limitations:**

Limitations are adequately addressed in the paper.

---

> ### Author Rebuttal · Authors · 2023-08-09
>
> Dear Reviewer MDXU,
>
> Thank you very much for your thoughtful feedback and comments! We are pleased to learn that you find our method to be **solid** and that we address an important problem through **extensive experiments**. We also appreciate your valuable suggestions and questions. In the following paragraphs, we have addressed the two mentioned weaknesses and four questions.
>
> ---
>
> ### W1: Missing two relevant works.
>
> We appreciate your bringing these two related papers to our attention. We will certainly include a discussion of these works in the next version of our paper. In brief, [1] and [2] have similar motivations to ours in developing human-like planning skills but differ in evaluation and methodology focus.
>
> ### W2: Statements about System 2 thinking.
>
> Thank you for the suggestion! We concur that the statement about System 2 requires clearer justification and clarification. Our primary intent was to highlight that SwiftSage is inspired by cognitive theory, and the analogy between System 1/2 and Swift/Sage modules might help readers grasp the concept more easily. We will revise the relevant paragraphs to address this concern appropriately.
>
> ### Q1: What about using classical planners as part of the solution?
>
> Indeed, classical planners could be applicable as well. However, they might necessitate more human-crafted design and conversion to specialized formal planning languages, and their performance is usually not on par with the three types of baseline methods we mentioned. In recent literature on agents for interactive tasks such as ALFWorld, Jericho, and ScienceWorld, classical planners are seldom implemented as baselines due to their limited generalizability during inference time.
>
> Nonetheless, we will revise our descriptions to ensure our claim is more inclusive and cite some classical planning methods in the related works for reference. We appreciate your question!
>
> ### Q2: Line 205. Typo of “SwiftSage”.
>
> You are correct; this is indeed a typo. The proper term should be “Sage” instead of “SwiftSage”. We will fix this in the next version. We appreciate you identifying the typo.
>
> ### Q3: What are the critical actions?
>
> In the ScienceWorld environment, we refer to all "focus on" actions as critical actions. Besides deciding the final answer, these actions also cover intermediate key decisions, such as confirming key objects (e.g., "focus on tin" when the task involves boiling tin). This condition can be customized based on the given environment and preferences. For example, critical actions may involve removing folders ("rm xxx") in script coding tasks like InterCode. Please refer to the newly uploaded pdf in the general rebuttal.
>
> Furthermore, we believe it holds potential to learn a dedicated module for classifying whether a time step is risky enough to activate Sage for additional planning. Hence, future work may focus on better generalizing the switching between the two modules.
>
> ### Q4: How much of the improvement is attributed to a balanced training set?
>
> We appreciate this insightful question. To address it, we have conducted new experiments for ablation analysis. When using the original, full data version, we find that the performance of Swift declines to 39.5. We also tested BART-large and smaller models, such as T5-base and T5-small, for similar comparisons. In all cases, utilizing task-balanced and action-balanced training trajectories results in better performance, increased training efficiency (i.e., obtaining a better model in less time), and smoother loss curves. T5-small exhibits a less significant difference in validation loss before and after balancing the data, we believe using balanced data. Overall, we think balancing the data in terms of task types and action types is a critical trick for imitation learning.
>
> ---
>
> Please do not hesitate to reach out with further questions, as we welcome continuing the discussion. If you feel that we have addressed your concerns sufficiently, would you kindly consider raising the rating score for our paper? Thank you very much in advance!
>
> Best regards,\
> Authors of SwiftSage

---

> > ### Comment · Reviewer_MDXU · 2023-08-17
> >
> > I thank the authors for the thoughtful response. I raised my rating to accept.

---

> > > ### Author Response · Authors · 2023-08-17
> > > **Thank you!**
> > >
> > > Dear Reviewer,
> > >
> > > We deeply appreciate your acknowledgment of our efforts and the raised score. We are also grateful for your valuable suggestions and will ensure their incorporation into our revisions.
> > >
> > > Best regards, \
> > > Authors of SwiftSage

---

### Official Review · Reviewer_Sgwx · 2023-07-04

**Soundness:** 4 excellent
**Presentation:** 4 excellent
**Contribution:** 4 excellent
**Rating:** 6
**Confidence:** 4

**Summary:**

The paper proposes SwiftSage that predicts a high-level plan by Swift and prompts LLMs to conduct each high-level subgoal by Sage.
Swift is trained by imitation learning for action planning as fast intuitive thinking and Sage utilizes LLMs for better generalizable planning and exception handling as deliberate thought processes.
The proposed method achieves strong performances on the ScienceWorld benchmark over the recent state-of-the-art.

**Strengths:**

- The paper is generally written well and easy to follow.
- Utilizing LLMs for exception handling seems reasonable.
- The overall two-staged framework (intuitive planning -> deliberate planning) is intuitive and sounds sensible.
- The proposed method achieves strong performance over prior work by large margins in most cases in the ScienceWorld benchmark.

**Weaknesses:**

- Swift adopts imitation learning to train an encoder-decoder (T5) model but I'm not sure why not just prompting LLMs (see Q1).
- The environment provides if the agent fails at an action, which might not be always available (see Q2).
- SwiftSage utilizes LLMs for exception handling but it is not clear if strategies for exception handling provided by LLMs are always useful and can be conducted by agents (Q3).
- The motivation of each condition of switching to Sage is missing (see Q4).

**Questions:**

- Q1: The Swift module fine-tunes a T5 model with oracle trajectories of training tasks.
But there is some work [14,A] that plans a sequence of actions by providing task-relevant examples to LLMs.
Is there any reason why the authors take imitation learning rather than prompting LLMs?
Although it is fine-tuning and this is quite efficient compared to training from scratch, it does require additional training.
In addition, wouldn't it be that planning with LLMs can generalize better than the imitation-learning models as we can leverage commonsense knowledge of LLMs?

- Q2: It seems that SwiftSage can perfectly recognize when and why it fails based on the feedback from the environment, which might enable Sage to address exceptions effectively.
Can the proposed method address cases where the failure feedbacks are incorrect?

- Q3: I'm not sure the proposed method can ensure that LLMs always provide helpful exception-handling strategies. If LLMs provide some strategies that the agent cannot conduct (e.g., due to the absence of objects), how can the agent address the exception? Note that this is not about the excitability of plans, which is addressed in the "Grounding stage" in Sec. 3.3.

- Q4: In Sec. 3.4, it seems missing the motivation of the conditions to switch to Sage. Why does each condition imply when to switch to Sage? It would solidify the switching conditions to include them.

References\
[A] Liang et al. Code as Policies: Language model programs for embodied control. ICRA, 2023.


**Limitations:**

The authors have adequately addressed the limitations.

---

> ### Author Rebuttal · Authors · 2023-08-09
>
> Dear Reviewer Sgwx,
>
> We greatly appreciate your thoughtful review and are pleased to learn that you find the SwfitSage well-motivated, yielding strong performance over prior work. We value your four insightful questions (Q1-Q4) regarding the weaknesses of our approach and have provided detailed responses to each below. Furthermore, a one-page pdf addressing these points is also available in our general rebuttal.
>
> ---
>
> ### Q1: Why not use prompting LLMs for the Swift module?
>
> **Swift** is designed to harness the capabilities of smaller language models (commonly understood as those with less than 1 billion parameters) to learn **environment-specific knowledge** from training trajectories while minimizing resource consumption during LLM inference. For these reasons, LLM prompting may not be optimal for Swift, as larger LLMs are typically more powerful, making their smaller counterparts (e.g., 7B models) less effective with prompting for planning. Additionally, in-context examples relevant to specific tasks can only cover a limited range of actions and scenarios, thus resulting in a lower performance.
>
> We do recognize the benefits of greater generalizability and enhanced **commonsense knowledge** offered by prompting LLMs, which is why the **Sage** module is employed to encapsulate these strengths. In contrast, Swift serves as an expert model, learning environment-specific knowledge from offline data. Thus, imitation learning is deemed a more efficient and conventional method for achieving this objective.
>
>
> ### Q2: Can SwiftSage address cases where the failure feedback is incorrect?
>
> Inaccurate failure feedback, particularly from humans, indeed poses a practical challenge, although we could not evaluate SwiftSage in such scenarios with ScienceWorld. If the failure feedback is inaccurate or unavailable, we argue that the agent can still transition to Sage for replanning, albeit less efficiently (for example, when no positive reward is received for K consecutive steps; similar to the situation in Q3 below).
>
> We appreciate your suggestion and we do plan to include an experiment in the next version of the paper by introducing noise to ScienceWorld's feedback to simulate a more realistic environment.
>
> ### Q3: What if the LLMs’ subgoals are not feasible?
>
> Upon receiving an action buffer containing grounded actions, the agent first attempts to execute each action sequentially. Any invalid action (e.g., resulting from missing objects) triggers a failure message, which is stored as an observation. Subsequently, these failure messages become part of the input (as observations in action history) for the next prompts.
>
> This enables the Sage module to infer that previous subgoals were infeasible in the current environment, allowing it to generate a refined plan in response to the 5th prompting question in the planning stage.
>
> ### Q4: What is the motivation for each condition in switching from Swift to Sage?
>
> We will include a clearer explanation of the motivation behind each condition in the revised Sec 3.4. A summary is provided below, and more further details are available in the newly uploaded pdf.
>
> Four triggering conditions (Line 254) are established to invoke the Sage module, each with their respective motivation:
>
> (1). Swift agents may perform actions that yield no rewards. Thus, activating Sage after K non-reward steps enables reflection and replanning. Also, for complex subgoals necessitating multiple steps, Sage can confirm progress (and/or intervene) with LLM prompting if necessary.
>
> (2). If predicted actions or those in the buffer are inexecutable (i.e., cannot be parsed by the ScienceWorld engine), it often indicates Swift's low confidence or inability to generalize to current out-of-distribution situations. Thus, we should use the Sage module that is engaged for its superior world knowledge for planning and reasoning.
>
> (3). Since the Swift module is generally weaker than Sage, the latter is activated for critical, potentially irreversible actions, such as providing answers in ScienceWorld tasks, submitting forms in web tasks, or removing files/folders in coding tasks. This criterion can be customized to suit the given environment and human preferences.
>
> (4). Analogous to the 2nd condition, when an unexpected observation surfaces following an executable action, the Sage module is prompted to replan based on these unforeseen factors (e.g., broken or missing required objects). Figure 2 exemplifies such a scenario.
>
> ---
>
> We welcome any further questions and are eager to continue the discussion. If our explanations have adequately addressed your concerns, may we kindly request that you consider raising your rating score for our paper? Thank you in advance!
>
> Best regards, \
> Authors of SwiftSage

---

> > ### Comment · Reviewer_Sgwx · 2023-08-16
> > **Rebuttal Acknowledgment**
> >
> > I thank the authors for addressing my concerns with a comprehensive explanation. The thorough response has clarified the issues I raised. I am happy to raise my rating accordingly.

---

> > > ### Author Response · Authors · 2023-08-17
> > > **Thank you!**
> > >
> > > Dear Reviewer Sgwx,
> > >
> > > We deeply appreciate your acknowledgment of our efforts and the raised score. We are also grateful for your valuable suggestions and feedbacks. We will incorporate them in the next version of the paper. Thanks again!
> > >
> > > Best regards, \
> > > Authors of SwiftSage

---

### Official Review · Reviewer_CSRK · 2023-07-06

**Soundness:** 4 excellent
**Presentation:** 4 excellent
**Contribution:** 4 excellent
**Rating:** 8
**Confidence:** 5

**Summary:**

This paper proposes a dual-module approach for ScienceWorld which combines an imitation learning agent based on t5, with an llm-based module which activates based on a heuristic to re-plan its actions, using a buffer of actions to be executed.
The approach is novel, sensible and well-motivated, drawing from the concept of slow-fast thinking from psychology.
The results significantly advance the state of the art on ScienceWorld, and significantly outperform the compared methods which include state of the art LLM methods like Reflexion.

**Strengths:**

- The paper significantly advances the state of the art on a difficult and interesting task involving language-based interaction, ScienceWorld. I expect that the proposed model will be a useful benchmark for many tasks of a similar type.
- The method is sound and well designed. Using seq2seq for a history-aware imitation learning is a sensible model for "fast thinking". And using an LLM for "deliberate thinking", i.e. object and progress tracking, planning and exception handling is also reasonable. Both components are well crafted, and they are nicely integrated in a complementary manner.
- The ablations are informative - Ablations show that the Swift module alone establishes a new state of the art, while GPT-4 almost doubles this performance. However chatgpt-turbo, which is comparatively much weaker than GPT-4, also improves Swift-only by about 25%, which is encouraging.

**Weaknesses:**

- I'm curious about the sensitivity of the method to prompt engineering. Q1~5 are quite comprehensive and capture a lot of the core aspects required to successfully solve the game. This is possible for ScienceWorld but would not be possible for many other tasks. Would it be possible to ablate the Qs to show prompt sensitivity?
- A weakness of the model is that it relies on a heuristic to decide when to activate the Sage module. I'm curious whether there are no better alternatives or more general solutions to the problem of integrating the two modules?

**Questions:**

See above.

**Limitations:**

Limitations discussion is adequate.

---

> ### Author Rebuttal · Authors · 2023-08-09
>
> Dear Reviewer CSRK,
>
> Thank you very much for your thoughtful feedback and comments! We are glad to hear that you believe our method is well designed and significantly advances the state of the art in complex interactive reasoning, as well as finding our evaluation informative. We also appreciate your concerns and suggestions.
>
> ---
>
> ### Q1: Would it be possible to ablate the Qs to show prompt sensitivity?
>
> We acknowledge that many other task types cannot directly utilize Qs in the current SwiftSage implementation. For example, there are no “objects” in some tasks, requiring customization of the Qs with task-specific prompts. Changes such as rephrasing Q1 and Q2 by replacing “objects” with task-specific elements are necessary. Q3-5, however, are largely task-general as they involve planning subgoals and self-reflection. We have included examples from other tasks, such as interactive coding and web-based tasks, in the updated PDF. For instance, Q1-2 will be about “files” in bash coding tasks, and “tables” for SQL tasks, etc.
>
> Regarding “ablate the Qs to show prompt sensitivity,” we had two interpretations: 1) maintaining the five questions with different wording and descriptions or 2) retaining only a subset of the questions. Due to the time constraints, we here provide some preliminary findings on a mini-size validation set, and we will provide a comprehensive report in the next version of the paper.
>
> 1) We have experimented with various phrasings for each question, finding that longer, more specific, and contextual descriptions are generally beneficial. For instance, we added sentences that set the context and invoked commonsense knowledge (e.g., “You are a teacher who instructs elementary students to do science experiments.” and “Please use commonsense knowledge when it is necessary.”) Removing such detailed descriptions results in a 4.3 point decrease in performance.  We also tried to merge the questions. Merging specific questions, like Q1 & Q2, led to a 2~3 point performance drop.
>
> 2) In ablation studies, we removed Q1+Q2 and Q5 to evaluate their importance (Q3 & Q4 are required for planning subgoals). Our results show that both are crucial, with a 9.6 point performance decrease for removing Q1+Q2 and a 13.5 point drop for removing Q5.
>
> ### Q2: Any alternatives for integrating Swift and Sage?
>
> Thank you for raising this point. We believe exploring alternative methods for integrating Swift and Sage is a promising future direction. One possible straightforward solution is to make Swift as a multi-task model that can also output some meta information about the states, such as if it is very uncertain about its predictions or the current action is rather risky such that it needs Sage to help. However, in order to learn such ability in either multi-task Swift or a dedicated separate module, we need supervision. One possible way to get such supervision can be from running the Swift agents in training variations and performing DAgger-style training. We will add more discussion on this topic in the next version of the paper. Thanks again for the great question.
>
> ---
>
> Please let us know if you have any further questions, and we are more than happy to continue the discussion.
>
>
> Thank you very much!
>
> Best regards, \
> Authors of SwiftSage

---

> > ### Author Response · Authors · 2023-08-18
> >
> > Dear Reviewer CSRK,
> >
> > Would you please read our rebuttal? We believe the rebuttal and the newly uploaded PDF should answer your questions.  If there are any outstanding issues, we would like the chance to respond before the discussion period is over.
> >
> > Thanks again for your thoughtful review!
> >
> > Best regards, \
> > Authors of SwiftSage

---

> > > ### Comment · Reviewer_CSRK · 2023-08-21
> > >
> > > I really appreciate the authors for the additional ablation experiments, and showing how the proposed method can be extended to different types of tasks. Keeping my rating of strong accept.

---

> > > > ### Author Response · Authors · 2023-08-21
> > > >
> > > > Thank you very much!

---

### Official Review · Reviewer_AXAQ · 2023-07-07

**Soundness:** 3 good
**Presentation:** 3 good
**Contribution:** 3 good
**Rating:** 6
**Confidence:** 4

**Summary:**

This paper proposes a new agent framework for action planning in complex interactive reasoning tasks. The authors leveraged an imitation-learning trained "light-weight" LLM for fast inference and an LLM (e.g. GPT4) for slow reasoning with proper designs of prompts and interaction between fast/slow modules. The proposed model achieve state-of-the-art results on ScienceWorld, outperforming existing models by a large margin.

**Strengths:**

- This idea of two-system in LLM is a new agent framework that could potentially be extended to many more settings and tasks.

- The proposed model outperforms existing baselines by a large margin on ScienceWorld which proves the effectiveness of the two-stage design.


**Weaknesses:**

- One concern about the current framework is its generalizability to more common tasks. It seems that the current design is strongly related to the ScienceWorld benchmark, it would be best to see if the current prompt/interaction designs could be still effective in customized settings, how to define the prompts for a general task, and also how sensitive they are to for example customized defined states, scores, etc.

- As 770M is already quite a big model before the LLM era, is the current Swift module capable of what we desire of accurate fast reasoning? This is reflected by the swift-only experiments that can already outperform existing RL/IL-based learning methods by a large margin. It makes one wonder to what extent can swift-module handle the reasoning problem, what are its failure cases, when does it need the Large LLM for slow reasoning.

**Questions:**

See the Weakness section.

**Limitations:**

The authors have properly addressed the limitations of their work.

---

> ### Author Rebuttal · Authors · 2023-08-09
>
> Dear Reviewer AXAQ,
>
> Thank you very much for your thoughtful feedback and comments! We are glad to hear that you believe our paper is well motivated and addresses the important problem of interactive reasoning and planning. We appreciate your concerns and have addressed your two questions below.
>
> ---
>
> ### Q1: Generalizability to more tasks; how to customize the prompts for general tasks.
>
> Thank you for this excellent question. In the newly uploaded one-page PDF, we present a general recipe for applying SwiftSage to other tasks, despite our focus on embodied tasks. The PDF demonstrates the customization of SwiftSage for coding tasks, web tasks, math problems, and complex Q&A.
>
> Notably, the first two task types have **interactive** evaluation environments (recently released) with straightforward observations and rewards. In contrast, the latter two receive limited feedback from existing benchmarking environments, and current datasets are unsuitable for testing agents in complex interactive tasks.
>
> We are conducting experiments on these benchmarks and will include the results in the next version. Please note that many interactive coding task environments (e.g., InterCode [1]) and web task benchmarks (e.g., WebArena [2]) were released after the NeurIPS deadline. We are working through some technical issues in their codebases and contacting their authors for assistance. Nonetheless, our one-page PDF demonstrates the general method of customizing SwiftSage for complex interactive tasks, and we hope it can answer your question.
>
> ### Q2: More error analysis for the Swift module; when does it need the Sage module.
>
> Thank you for raising this important question. We previously investigated failure cases for Swift-only experiments. Due to space limitations, we cannot provide full examples here. Instead, we summarize our findings below.
>
> (1). Swift may generate invalid actions, necessitating the choice of one for execution. In preliminary experiments, we tested various decoding algorithms, all of which often generated invalid actions unparseable by ScienceWorld. Without the Sage module, we must either choose less confident yet valid actions or select the most similar valid action from a pool if all top candidates are invalid. In essence, if Swift is unconfident in its predictions or cannot generalize in the current situation, more risky actions must be executed, which can propagate mistakes to future steps.
>
> (2). Swift cannot replan when mistakes occur or exceptions are encountered, often leaving it lost in later time steps. This issue is partly due to Swift's limited context window for memorizing action history. In contrast, Sage can effectively reflect on action history and replan subsequent subgoals and actions, a skill Swift is not trained to perform.
>
> (3). Swift tends to make decisions before identifying correct objects. As a greedy algorithm trained with imitation learning, the Swift module aims to complete tasks quickly. However, it can overlook the absence of necessary objects or proper conditions, potentially leading to incorrect actions and missed score points.
>
> (4). Swift has limited commonsense knowledge. Some task variations require reasoning about object locations, affordances, animal lifespans, color mixtures, etc.
>
> These scenarios demonstrate the need for Sage and why SwiftSage significantly outperforms the Swift-only module. Our motivation for the four conditions activating Sage stems from these observations. We will include these findings with more concrete examples in the next version.
>
> ---
>
> Please let us know if you have any further questions, as we are happy to continue the discussion. If you find that our response addresses your concerns, would you kindly consider raising your rating score for our paper? We greatly appreciate your consideration.
>
> Best regards,\
> Authors of SwiftSage

---

> > ### Author Response · Authors · 2023-08-18
> >
> > Dear Reviewer AXAQ,
> >
> > Would you please read our rebuttal? If there are any outstanding issues, we would like the chance to respond before the discussion period is over.
> >
> > We believe the rebuttal and the newly uploaded PDF should answer your questions and may merit an increase in score if you also think they are helpful.
> >
> > Thanks again for your thoughtful review!
> >
> > Best regards, \
> > Authors of SwiftSage

---

> > > ### Comment · Reviewer_AXAQ · 2023-08-21
> > > **Post-rebuttal response**
> > >
> > > The authors have addressed most of my concerns, therefore I'm keeping my original rating.

---

> > > > ### Author Response · Authors · 2023-08-21
> > > >
> > > > Thank you very much!

---

### Author Rebuttal · Authors · 2023-08-09

### General response

Dear Reviewers and AC,

We greatly appreciate your insightful reviews and are delighted that all of you have acknowledged our paper's positive aspects. We briefly summarize the favorable feedback as follows:

- **Well-motivated, innovative, generalizable method**: Sgwx, MDXU, CSRK, AXAQ;
- **Significant performance improvements**: Sgwx, AXAQ, CSRK, MDXU;
- **Extensive evaluation & informative ablation**:  MDXU, CSRK;
- **Well-written and easy to follow**: Sgwx, MDXU, CSRK;

---

We sincerely thank the reviewers for their constructive suggestions and questions to enhance our paper. We trust that our rebuttal has sufficiently addressed the concerns raised by the reviewers. Please do not hesitate to contact us if you have any further questions, and we will be more than happy to continue the discussion.

---
Aside from the rebuttal text, we are also providing **a one-page pdf** that includes the following content:

(1). A pseudocode to better illustrate the SwiftSage framework, which we hope clarifies some details of our framework.

(2). A set of figures demonstrating SwiftSage's customization for other complex interactive tasks such as coding tasks, web tasks, math problems, and complex QA. We believe this will help showcase SwiftSage's potential generalization and serve as a general recipe for extending SwiftSage to other applications.

Once again, thank you for your valuable input. If our responses have addressed your concerns, please kindly consider increasing your scores. We sincerely appreciate your consideration.

Best regards, \
Authors of SwiftSage

---

### Decision · Program_Chairs · 2023-09-21

**Decision:**

Accept (spotlight)

**Comment:**

This work proposes a two-stage method for action planning in complex interactive reasoning tasks: a light-weight model for fast intuitive thinking, and a more complex LLM for subgoal planning and grounding. The proposed method is shown to achieve STOA results on ScienceWorld.

Reviewers recognize the new framework, significant improvements over existing methods as well as the extensive experiments.

The authors addressed the reviewers concerns quite extensively in the rebuttal. Two of the reviewers raised their scores, and the final ratings are unanimously accept (6, 8, 6, 7).

Overall this is a solid and timely work to show that complex reasoning tasks can be better solved with a two-stage approach, similar to slow-fast thinking from psychology.

Please add the additional experiments/discussions (especially the ablation experiments and the discussions regarding how the method can be extended to more general settings) in the final version.